# A Whole-Genome Analysis of the African Swine Fever Virus That Circulated during the First Outbreak in Vietnam in 2019 and Subsequently in 2022

**DOI:** 10.3390/v15091945

**Published:** 2023-09-18

**Authors:** Van Phan Le, Min-Ju Ahn, Jun-Seob Kim, Min-Chul Jung, Sun-Woo Yoon, Thi Bich Ngoc Trinh, Thi Ngoc Le, Hye Kwon Kim, Jung-Ah Kang, Jong-Woo Lim, Minjoo Yeom, Woonsung Na, Xing Xie, Zhixin Feng, Daesub Song, Dae Gwin Jeong

**Affiliations:** 1Department of Microbiology and Infectious Diseases, College of Veterinary Medicine, Vietnam National University of Agriculture, Hanoi 100000, Vietnam; letranphan@vnua.edu.vn (V.P.L.); trinhthibichngoc.vet@gmail.com (T.B.N.T.); 2Department of Proteome Structural Biology, KRIBB School of Bioscience, University of Science and Technology, Daejeon 34141, Republic of Korea; amj315@kribb.re.kr (M.-J.A.); minchul@kribb.re.kr (M.-C.J.); 3Bionanotechnology Research Center, Korea Research Institute of Bioscience and Biotechnology, Daejeon 34141, Republic of Korea; ngoc@kribb.re.kr (T.N.L.); kjungah@kribb.re.kr (J.-A.K.); 4Department of Nano-Bioengineering, Incheon National University, Incheon 22012, Republic of Korea; junkim@inu.ac.kr; 5Department of Biological Science and Biotechnology, Andong National University, Andong 36729, Republic of Korea; syoon@andong.ac.kr; 6Department of Microbiology, College of Natural Sciences, Chungbuk National University, Cheongju 28644, Republic of Korea; khk1329@chungbuk.ac.kr; 7College of Veterinary Medicine and Research Institute for Veterinary Science, Seoul National University, Seoul 08826, Republic of Korea; nanobiolim@snu.ac.kr (J.-W.L.); virusxnox@snu.ac.kr (M.Y.); 8College of Veterinary Medicine, Chonnam National University, Gwangju 61186, Republic of Korea; wsungna@jnu.ac.kr; 9Key Laboratory for Veterinary Bio-Product Engineering, Ministry of Agriculture and Rural Affairs, Institute of Veterinary Medicine, Jiangsu Academy of Agricultural Sciences, Nanjing 210014, China; xiexing@jaas.ac.cn (X.X.); fzxjaas@163.com (Z.F.)

**Keywords:** African swine fever, next-generation sequencing, complete genome sequence, virus evolution

## Abstract

Since its initial report in Vietnam in early 2019, the African swine fever (ASF), a highly lethal and severe viral swine disease worldwide, continues to cause outbreaks in other Southeast Asian countries. This study analyzed and compared the genomic sequences of ASF viruses (ASFVs) during the first outbreak in Hung Yen (VN/HY/2019-ASFV1) and Quynh Phu provinces (VN/QP/2019-ASFV1) in Vietnam in 2019, and the subsequent outbreak in Hung Yen (VN/HY/2022-ASFV2) in 2022, to those of other ASFV strains. VN/HY/2019-ASFV1, VN/QP/2019-ASFV1, and VN/HY/2022-ASFV2 genomes were 189,113, 189,081, and 189,607 bp in length, encoding 196, 196, and 203 open reading frames (ORFs), respectively. VN/HY/2019-ASFV1 and VN/QP/2019-ASFV1 shared a 99.91–99.99% average nucleotide identity with genotype II strains. Variations were identified in 28 ORFs in VN/HY/2019-ASFV1 and VN/QP/2019-ASFV1 compared to 20 ASFV strains, and 16 ORFs in VN/HY/2022-ASFV2 compared to VN/HY/2019-ASFV1 and VN/QP/2019-ASFV1. Vietnamese ASFV genomes were classified as IGR II variants between the I73R and I329L genes, with two copy tandem repeats between the A179L and A137R genes. A phylogenetic analysis based on the whole genomes of 27 ASFV strains indicated that the Vietnamese ASFV strains are genetically related to Estonia 2014, ASFV-SY18, and Russia/Odintsovo_02/14. These results reveal the complete genome sequences of ASFV circulating during the first outbreak in 2019, providing important insights into understanding the evolution, transmission, and genetic variation of ASFV in Vietnam.

## 1. Introduction

African swine fever (ASF) is a serious transboundary disease and highly infective among domestic pigs and wild boars. The ASF virus (ASFV, family *Asfaviridae*) causes economic damage to farms due to stamping-out, social disruption due to a high mortality and quarantine, and pork import and export regulations [1]. ASFV is a large double-stranded DNA virus with a total genome size of approximately 175,000–195,000 base pairs (bp) encoding approximately 150 to over 200 open reading frames (ORFs) [2,3,4]. ASF is endemic in Sub-Saharan Africa and is transmitted between warthogs and bushpigs with soft ticks. ASFV can also be transmitted among pigs through direct contact with infected pigs, their blood, meat, or other tissues, indirect contact with contaminated objects, or via the air over short distances [4]. ASFV strains isolated from various countries have been classified into 24 genotypes through the comparative analysis of B646L (p72 genotype) partial gene sequences [5]. As only a limited number of live attenuated ASF vaccines have been commercialized and are not yet widely available, the effective prevention of ASF relies on the culling of pigs and regulation of internationally traded pork products through early detection of ASFV spread into new countries [2].

In 1921, ASF was first reported in Kenya (East Africa) as a viral hemorrhagic fever in domestic pigs; however, by the 1950s, it had spread to most African countries [6,7,8]. In 1957, ASF was reported as spreading intercontinentally, from Africa to Portugal (Europe). Over the following decades, it spread to other European and South American countries [9]. In 2007, the third international outbreak of ASF occurred in Georgia and was associated with the p72 genotype II strain, the most isolated worldwide [4]. ASFV spread throughout Russia and Eastern Europe [4,10,11,12], and from 2012 to 2018, the p72 genotype II strain affected Eastern and Central Europe; from 2014 to 2018, ASF was reported across Europe [4,10,13,14]. Recently, many ASF cases in Europe have been consistently reported in Poland, Portugal, Belgium, and Italy between 2022 and 2023. In 2023, over 100,000 cases of ASFV were reported in Poland, and over 1 million pigs were slaughtered. ASFV was reintroduced in Portugal in 2022, and over 1,000 cases have been reported since then. ASFV was first reported in Belgium in December 2022; more than 100 cases of ASFV were reported in wild boars in several country regions. No cases of ASFV were recently reported in domestic pigs in Portugal and Belgium. Moreover, ASFV was first reported in Italy in January 2023; more than 50 cases were reported, and ASF was still spreading [15]. In Asia, the first outbreak of ASF occurred in Northeast China in August 2018 (city of Shenyang, province of Liaoning) [4,16]. Shortly following this outbreak, ASFV was isolated and reported in Vietnam in February 2019, and subsequent ASF outbreaks were reported in Southeast Asia, including Mongolia [13,17].

Before the official report of the ASFV outbreak in Vietnam in 2019, clinical symptoms similar to ASF had occurred and were reported in all major pig farms in the northern and southern regions of Vietnam. Based on the temporal analysis of outbreak regions in China and considering the potential introduction of ASFV from the border regions of China, several ASF outbreaks occurred in the Yunnan and Guangxi provinces bordering Vietnam in southwestern China from October to December 2018 [18]. Since the first outbreak in Vietnam, many studies have been performed on the genetic variation of ASFV [19,20,21,22]. Recently, several research groups in Vietnam have reported next-generation sequencing (NGS) results of ASFV isolated in 2020 and 2021 from the northern and southern regions of Vietnam [20,21]. However, accurately explaining the evolution and diversity of ASFV in Vietnam has been challenging owing to the absence of complete genomic information on the first ASFV strain occurring in Vietnam.

In this study, we collected ASFV-positive domestic pig tissues during the first 2019 outbreaks in Hung Yen and Quynh Phu provinces, northern Vietnam and whole blood samples in 2022 in Hung Yen and conducted NGS on these samples. We assembled the complete coding regions of the ASFV genomes from 2019 and 2022 in Vietnam. Notably, we obtained whole genomic sequences during the first outbreak in Vietnam directly from spleen tissues and whole blood samples without ASFV propagation and enrichment through a cell culture. Moreover, we compared the genome of the Vietnamese strain to those of the China/2018/AnhuiXCGQ and related European p72 genotype II strains. Our results provide important insights into the evolution of ASFV following its transboundary spread and essential genetic information for developing vaccines, and diagnostic markers, and implementing effective control measures to prevent the spread of ASF.

## 2. Materials and Methods

### 2.1. Sample Collection

ASFV-infected spleen tissue samples were collected in 21 breeding sows from a pig farm in Hung Yen (VN/HY/2019-ASFV1) and 24 fattening pigs in Quynh Phu (province) (VN/QP/2019-ASFV1) in northern Vietnam. The pigs displayed several distressing symptoms, including a sudden loss of appetite, high fever, anorexia, vomiting, lethargy, respiratory distress, coughing, instances of abortion, and disseminated cyanosis, and the first to show ultimately died 3 days later in the first outbreak in 2019 [19,23]. A total of 5 recent whole blood samples were collected from ASFV-infected sows on a farm in Hung Yen in November 2022 (VN/HY/2022-ASFV2). ASFV was detected using a commercial real-time polymerase chain reaction (PCR) kit, the VDx^®^ ASFV qPCR kit (Median Diagnostics Inc., Chuncheon, Gangwon-do, South Korea), that amplifies a target sequence in the ASF viral B646L (p72) genome. A total of 10 mg of individual spleen tissues and 200 µL of whole blood obtained from deceased pigs were homogenized with a grinding process in phosphate-buffered saline (PBS) and lysed using the addition of Proteinase K, heated at 56 °C for up to 1 h, and centrifuged at 12,000× *g* for 10 min. A total of 200 μL of the lysate supernatant was used for viral DNA extraction with a QIAamp Viral DNA Mini Kit 250 (Qiagen, Hilden, Germany) [19].

### 2.2. Library Preparation

Library preparation was performed according to the TruSeq Nano DNA Library Prep Guide (Illumina Inc., San Diego, CA, USA). In brief, 100 ng of 3 genomic DNA samples from each VN/HY/2019-ASFV1, VN/QP/2019-ASFV1, and VN/HY/2022-ASFV2 sample was sheared using an LE220 Focused-ultrasonicator (Covaris Inc., Woburn, MA, USA) with a duty factor of 15% and peak incident power of 450 W at 200 cycles per burst for 50 s. Sheared DNA fragments were then end-repaired, size-selected to obtain DNA fragments of approximately 350 bp, and adenylated according to the manufacturer’s instructions. After ligating index adapters to the ends of the DNA fragments, DNA libraries were enriched using eight cycles of PCR, according to the manufacturer’s instructions. After size selection and PCR amplification, the quality of the library and band size were assessed using D1000 Screen Tapes on a Tapestation 2200 (Agilent Technologies, Santa Clara, CA, USA). Finally, libraries were quantified using the PicoGreen dsDNA quantitation assay (Thermo Fisher Scientific, Waltham, MA, USA) and determined using a Victor3 plate reader (PerkinElmer, Waltham, MA, USA).

### 2.3. Clustering and Sequencing

Illumina Inc. uses a unique bridged amplification reaction on the surface of a flow cell. A flow cell containing the DNA libraries was prepared using the cBot fluidics station and then loaded onto the HiSeq™ 4000 platform (Illumina Inc.) for automated cycles of extension and imaging. The sequencing-by-synthesis cycle was repeated for a paired-end read length of 2 × 100 bp.

### 2.4. Genome Sequencing and Annotation

The whole genomes of VN/HY/2019-ASFV1, VN/QP/2019-ASFV1, and VN/HY/2022-ASFV2 were sequenced using the Illumina HiSeq™ 4000 platform. First, sequence reads were assembled using SPAdes to generate a single-cell assembler [24]. Each VN/HY/2019-ASFV1, VN/QP/2019-ASFV1, and VN/HY/2022-ASFV2 strain yielded two contigs that aligned to the genome sequence of the China/2018/AnhuiXCGQ strain, which was the first isolated and reported strain in Asia in August 2018 and was assumed to have high similarity to Vietnam strains using the software MEGA5. The contig gaps were resolved by sequencing the PCR products over the gaps, and the open reading frames (ORFs) of the assembled genome were predicted using Rapid Annotation on a Subsystem Technology Sever. Finally, the predicted ORFs were manually curated using the ORF prediction results of GeneMarkS [25,26] and FgenesV0 [27].

### 2.5. Comparative Genomic Analysis

The CLC Genomics Workbench 22.0.2 (QIAGEN) was used for a comparative genomic analysis. For comprehensive genome comparison, 25 ASFV strains were selectively isolated as the first outbreak and reported in several countries by 2019. First, the whole-genome average nucleotide identity (ANI) and alignment percentage (AP) were computed using the ANI 1.0 module [28]. In addition, the nucleotide and amino acid sequences of 28 significant ORFs were compared using heatmaps and dendrograms, which were created using the Tidyverse and heatmap packages in R (v4.1.3) [29,30]. Pairwise multiple sequence alignments of the intergenic regions (IGRs) of 28 ASFV strains were performed using MUSCLE software 3.0 [31]. Finally, 27 ASFV whole-genome sequences, including the Vietnam strains, were aligned using the Alignment 1.02 module, and a phylogenetic tree was constructed using the maximum likelihood phylogeny 1.3 module [32].

## 3. Results

### 3.1. Preparation of ASFV-Infected Samples

In 2019, during the first outbreak in Vietnam, the pigs of sows in Hung Yen and Quynh Phu province pig farms showed the first clinical signs of the disease, including high fever, anorexia, vomiting, lethargy, respiratory distress, coughing, abortion, and unwillingness to stand, and the first sow died 3 days later. ASFV was detected, using commercial real-time PCR targeting of the B646L (p72) gene, in 21 of the 33 dead pigs in Hung Yen in February 2019 and all 24 pigs in Quynh Phu (province) in May 2019. After the first outbreak in Vietnam, ASFV continued to occur and ASFV-positive whole blood samples were collected in Hung Yen from five dead pigs in November 2022. We collected three genomic DNA samples at various time points (2019–2022) and from different regions in Vietnam (Hung Yen and Quynh Phu provinces). We conducted whole-genome sequencing on the collected samples from Hung Yen and Quynh Phu in 2019 and Hung Yen in 2022, with cycle threshold (Ct) values of 26.94, 13.58, and 15.1 using real-time PCR, respectively.

### 3.2. Complete Genome Sequence Analysis of VN/HY/2019-ASFV1 and VN/QP/2019-ASFV1

After whole-genome sequencing, we successfully analyzed the VN/HY/2019-ASFV1 and VN/QP/2019-ASFV1 strains circulated in Hung Yen and Quynh Phu in 2019, respectively. Two contigs of each Vietnamese strain were obtained using de Devo SPAdes assembly software, and 196 ORFs were identified using the RAST server. The complete genome sequences of VN/HY/2019-ASFV1 and VN/QP/2019-ASFV1 were 189,113 and 189,081 bp in length, respectively. Compared with the ASFV China/2018/AnhuiXCGQ genome, which represents the first ASFV outbreak in China in 2018, both sequences showed deletion of nucleotides, spanning from position 420 to 810 at the 5′ end. Compared with the VN/HY/2019-ASFV1 genomic sequence, that of VN/QP/2019-ASFV1 showed a deletion of 37 bp at the 3’ end. The pairwise comparison showed six variable nucleotide sites between VN/HY/2019-ASFV1 and VN/QP/2019-ASFV1, including four deletion and two insertion sites. However, these two genome sequences showed 100% ANI, even with these deletion and insertion sites (Appendix A).

### 3.3. Genomic Comparison of VN/HY/2019-ASFV1 and VN/QP/2019-ASFV1 with Genotype I and II ASFV Strains

We first reported the partial genome sequence of B646L (p72), EP402R (CD2v), and E183L (p54) in Vietnam in 2019. These sequences were classified into p72 genotype II and CD2v serotype 8, which are identical to the Georgia 2007/1 and China/2018/AnhuiXCGQ strains [17]. The GenBank accession number, genome size, genotype, country, and year of isolation of each of these sequences for the ASFV genomes are listed in Appendix A. The ORFs identified in VN/HY/2019-ASFV1 and VN/QP/2019-ASFV1 were compared to those of other ASFV strains, and their sequence differences were analyzed with pairwise sequence alignment. The VN/HY/2019-ASFV1 and VN/QP/2019-ASFV1 genomes shared 99.91–99.99% ANI with p72 genotype II and 96.86–97.32% ANI with p72 genotype I. The VN/HY/2019-ASFV1 and VN/QP/2019-ASFV1 genomes shared 99.93% ANI with China/2018/AnhuiXCGQ, ASFV/LT/1490, Georgia 2007/1, and POL16_20186_o7, and 99.99% ANI with Estonia 2014 (Appendix A).

As shown in Figure 1, we identified 28 ORFs, namely C315R, C717R, C84L, CP123L, CP204L, D1133L, DP238L, E199L, EP153R, I96L, I267L, 173 L, I73R, K145R, KP177R, L11L, M1249L, multigene family (MGF)_110-13L, MGF_110-13Lb, MGF_110-14L, MGF_110-1L, MGF_360-14L, MGF_360-16R, MGF_360-1L, MGF_360-21R, MGF_505-5R, MGF_505-9R, NP419L, and O174L. These ORFs exhibited variations compared to the conservative ORF sequence of the Vietnamese strains, differing by at least one base pair or amino acid compared to 20 of the 25 ASFV strains analyzed in this study. We compared them with only 28 ORF sequences of 20 ASFV strains, except for 5 ASFV strains with 100% identity, using multiple sequence alignment. The heatmap color spectrum reflects the strength of the identity (the warmer the color, the higher the identity, and the cooler the color, the lower the identity). Compared to the EP153R ORF of VN/HY/2019-ASFV1 and VN/QP/2019-ASFV1, that of the Spain/E75, Portugal/L60, Benin 97/1, Italy/47/Ss/2008, Italy/26544/OG10, Spain/BA71, Portugal/OURT 88/3, and Portugal/NHV/1968 sequences had 2 deletions, 3 insertions, and 45 amino acid point mutations (Appendix A).

Most of the p72 genotype I strains with variable MGFs, including MGF360, MGF110, and MGF505, exhibited more modifications and gene deletions than p72 genotype II, which includes VN/HY/2019-ASFV1 and VN/QP/2019-ASFV1. Comparing the MGF360-1L, MGF110-14L, and KP177R genes of VN/HY-ASFV1 with the Estonia 2014 genomic sequence, the genomic sequence of VN/HY-ASFV1 shared 10.32–89.26% nucleotide and 40.62–83.2% protein identity with some of the genes missing from the Estonia 2014 strain (Appendix A). Comparing the MGF360-1L gene of VN/HY/2019-ASFV1 and VN/QP/2019-ASFV1 with the genome sequences of genotype I and II strains, 293 nucleotides were deleted at the 5′ end of MGF360-1L of genotype I. Eleven nucleotides were also remarkably deleted at the 3′ end of p72 genotypes I and II, except for ASFV-wBS01, CN_2019_InnerMongolia-AES01, and Georgia_2007_1, which contained more gene deletion regions (Appendix A).

In comparing the sequences of MGF110-1L of VN/HY-ASFV1 and VN/QP-ASFV1 with the genome sequences of p72 genotype I, 64 mutations were identified, and three nucleotides were deleted at positions 316–318. Moreover, at the 3′ end, 498 nucleotides were deleted in p72 genotype II, including VN/HY/2019-ASFV1 and VN/QP/2019-ASFV1, except for ASFV-SY18, ASFVL14, and Georgia 2007_1, which had a deletion of 170 nucleotides (Appendix A). The 5′ and 3′ end MGFs of VN/HY/2019-ASFV1 and VN/QP/2019-ASFV1, specifically MGF360, MGF110, MGF505, MGF300, and MGF100, were identical to those of the China/2018/AnhuiXCGQ strain. However, the Estonia 2014 strain had a relatively higher ANI with VN/HY/2019-ASFV1 and VN/QP/2019-ASFV1 than other ASFV p72 genotype II strains; the ORFs of MGFs and KP177R shared a low sequence identity due to gene insertions and deletions. In addition, both VN/HY/2019-ASFV1 and VN/QP/2019-ASFV1 strains shared ORFs with the China/2018/AnhuiXCGQ, P.O.L./2015/Podlaskie, and Georgia 2007/1 strains, including other Chinese ASFV strains reported since 2018, which have a higher sequence identity than the different p72 genotype II strains (Appendix A).

### 3.4. Genome Comparison of VN/HY/2022-ASFV2 with VN/HY/2019-ASFV1 and VN/QP/2019-ASFV1

The complete genome sequence of VN/HY2022-ASFV2 encodes 203 ORFs and 189,607 bp in length, including approximately 500 additional nucleotides at the 3′ end, with 99.98% ANI compared to VN/HY/2019-ASF1 and VN/QP/2019-ASF1. The ORFs that differed between VN/HY2022-ASFV2 and VN/HY/2019-ASF1 were ASFV_Ch_ACD_00290 QDL88036.1, ASFV_Ch_ACD_00290 QED21566.1, F8221_gp208, EP402R, and H233R. ASFV_Ch_ACD_00290 QDL88036.1 was only encoded in VN/HY/2019-ASF1 and VN/QP/2019-ASF1. EP402R and H233R, which were divided into four and two ORFs, respectively, were only detected in VN/HY2022-ASFV2. By comparing VN/HY2022-ASFV2 with VN/HY/2019-ASF1 and VN/QP/2019-ASF1, we detected 16 ORFs (MGF_110-3L, MGF_110-4L, MGF_110-5L-6L, MGF_110-14L, ASFV_G_ACD_00350, EP402R, NP419L, D339L, P1192R, H233R, E199L, E248R, I196L, MGF_360-16R, MGF_360-18R, and MGF_360-21R) showing nucleotide mutation, insertion, and deletion sites, and prominently identified 19 nucleotides’ mutation in MGF_360-21R, 66 nucleotides’ insertion in I196L, 11 nucleotides’ deletion in H233R, and 9 and 3 nucleotides’ deletion in MGF_110-14L of VN/HY/2019-ASF1 and VN/QP/2019-ASF1, respectively.

### 3.5. Sequence Alignment Analysis of the IGR in ASFV Genomes

Based on the IGR between the I73R and I329L genes, ten nucleotide tandem repeats of GGAATATATA were considered molecular fingerprints or genetic markers that are highly variable, specific, and easy to amplify. The four IGR repeats were classified as IGR I (two copies), II (three copies), III (four copies), and IV (five copies), depending on the number of tandem repeats [33]. The IGR analysis of VN/HY/2019-ASFV1, VN/QP/2019-ASFV1, and VN/HY/2022-ASF2 between I73R and I329L revealed a belonging to IGR II, similar to the China/2018/AnhuiXCGQ, Estonia 2014, Pol16_20186_o7, and other p72 genotype II strains (Figure 2a). We recently confirmed and reported the IGR III variant in the Vietnamese strains isolated from 2020 to 2021 [19]. Moreover, the analysis of the IGR between the A179L and A137R genes showed two or three copies of eleven nucleotide tandem GATACAATTGT repeats in p72 genotypes I and II. Pairwise sequence alignments showed that the IGRs of VN/HY/2019-ASFV1, VN/QP/2019-ASFV1, and VN/HY/2022-ASF2 located between A179L and A137R had two copies of tandem repeats, and were clustered with the IGR of p72 genotype II, unlike p72 genotype I, which had three copies of tandem repeats, except for SA/Mkuzi1979 (Figure 2b). The ASFV/VNPig/Hanoi/07 strain, which has one copy of a tandem repeat, was recently reported to have more deletions in the 11 nucleotide tandem repeats within the IGR between A179L and A137R than VN/HY/2019-ASFV1, VN/QP/2019-ASFV1, VN/HY/2022-ASF2, China/AnhuiXCGQ, and other p72 genotype II strains [34].

### 3.6. Phylogenetic Analysis

Sequence alignment was performed for the whole-genome sequences of the ASFV strains downloaded from NCBI using the CLC Genomics Workbench. The phylogenetic analysis using the maximum likelihood method revealed that all 27 ASFV strains could be divided into two p72 genotypes (Figure 3). VN/HY/2019-ASFV1 and VN/QP/2019-ASFV1 strains closely correlated with the Estonia 2014, ASFV-SY18, ASFV/LT/1490, Russia/Odintsovo_02/14, Pol/2015/Poldiaskie, and p72 genotype II strains. Although the Chinese ASFV strain had a high sequence similarity (99.93% ANI) with the Vietnamese strains, the Anhui and Wuhan strains were less closely related to the Vietnamese strains than European countries such as Estonia, Poland, and Lithuania, except for ASFV-SY18 (Chinese strain).

## 4. Discussion

In this study, we analyzed the first outbreak cases of Vietnamese ASFV strains circulating among northern pig farms in Hung Yen in February and Quynh Phu, Thai Binh in May 2019, and later in Hung Yen in November 2022. Notably, we used ASFV-infected pig spleen tissue and whole blood samples without the need for sub-culturing the virus. When the first ASFV outbreak was reported in Vietnam, we reported that ASFV might have spread from the first ASFV strain named China/2018/AnhuiXCGQ, which was detected in China in August 2018, sharing the same p72 genotype II and CD2v serotype 8 [17]. In the present study, our data revealed that the VN/HY/2019-ASFV1 and VN/QP/2019-ASFV strains shared a 99.93% ANI with China/2018/AnhuiXCGQ, including nucleotide deletions at the 5′ end, and over 99.91% ANI with other p72 genotype II strains. In addition to the China/2018/AnhuiXCGQ strain, the VN/HY/2019-ASFV1 and VN/QP/2019-ASFV strains exhibited high similarities with other strains, including Estonia 2014 (99.99% ANI), Georgia 2007/1 (99.93% ANI), POL16_20186_o7 (99.93% ANI), and ASFV/LT/1490 (99.93% ANI) strains. These findings suggest that ASFV in Vietnam might share a common origin with the Central and Eastern Europe ASFV strains [35,36].

In the present study, the results of the multiple sequence alignment analysis for 28 ORFs, particularly EP153R, I196L, KP177R, and MGFs of VN/HY/2019-ASFV1 and VN/QP/2019-ASFV1, showed distinct deletion regions compared to Estonia 2014 and the p72 genotype I strains. Mutations observed in EP153R, I196L, and KP177R are associated with increased virulence and transmissivity, potentially rendering the ASFV less susceptible to vaccines [37,38,39]. Moreover, three MGFs (MGF110, MGF360, and MGF505) of VN/HY/2019-ASFV1 and VN/QP/2019-ASFV1 displayed the most similarity at the 5′ and 3′ ends of the variable region with the p72 genotype II strains and showed more modifications and gene deletions than p72 genotype I strains. MGF360 and MGF505 genes are expressed in cells during early infection and are essential for determining host targeting and early host cell death following infection [40,41]. Compared to a highly pathogenic ASFV strain, the non-pathogenic isolate lacked eight MGF genes (MGF505-1R and MGF360-9L, 10L, 11L, 12L, 13L, 14L, and 2R), suggesting that MGF505 and MGF360 may play important roles in virulence and immune evasion [42]. The recent VN/HY2022-ASFV2 strain had a 99.98% ANI compared to VN/HY/2019-ASFV1 and VN/QP/2019-ASFV1 and showed variations in 16 ORFs, particularly with numerous mutations in MGF360-21R, I196L, and H233R ORFs, which are involved in virulence and transmission [38,43]. However, it remains unclear whether the diversity observed in EP153R, I196L, KP177R, I996L, H233R, and MGF in Vietnamese strains affects the virulence, viral transmissibility, and susceptibility to vaccines of the currently circulating Vietnamese ASFV, warranting further investigations.

In addition, the IGRs located between the I73R and I329L of VN/HY/2019-ASFV1, VN/QP/2019-ASFV1, and VN/HY/2022-ASF2 were classified as IGR II in p72 genotype II, which differs from IGR I in p72 genotype I. IGR II was the most common variant circulating in Vietnam, and IGR I was detected only in Hanoi and northern Vietnam in 2019. Recently, we detected four IGR III variants in three different regions of northern Vietnam; however, IGR III variants are not common in Vietnam and have been previously reported in China. Therefore, IGR III variants detected from 2020 to 2021 in northern Vietnam, near the Chinese border, are assumed to be the result of the transmission of ASFV from China [19]. VN/HY/2019-ASFV1, VN/QP/2019-ASFV1, VN/HY/2022-ASF2, and p72 genotype II contain two copies of tandem repeats in the I73R and I329L IGR. In contrast, ASFV/VNPig/Hanoi/07, which was recently identified as a novel variant, has one copy of this tandem repeat. Given the circulation of several IGR variants in Vietnam and ongoing reports of new cases, it is essential to understand and monitor the continuous emergence of IGR variants [33,34].

In the present study, the phylogenic analysis of 27 ASFV strains revealed that Vietnamese strains had a closer genetic relationship with Estonia 2014, ASFV-SY18ASFV/LT/1490, Pol/2015/Poldiaskie, and Russia/Odintsovo_02/14 than China/2018/AnhuiXCGQ in the p72 genotype II group. The Estonia 2014 strain first occurred in Estonia in 2014 and quickly spread to neighboring countries. ASF outbreaks occurred sequentially in Latvia, Lithuania, Poland, and Russia in 2015 and in Belarus, the Czech Republic, Slovakia, and Ukraine in 2016 [15]. The northern Vietnamese province shares a 1281 km border with China, and illegal transport of pigs and pig products frequently occurs in these border regions. However, the exact route and the time of the incursion of ASFV from China to Vietnam remain unknown [36]. During the first half of 2018, the General Department of Vietnam Customs reported that Vietnam imported thousands of tons of pork from Poland, the largest supplier. The Office International des Epizooties announced an ASF outbreak in Hungary and Poland from early 2018 to September 2018, resulting in infections and the culling of hundreds of wild boars and domestic pigs [13,14]. In response to the ASF threat, the Ministry of Agriculture and Rural Development decided to cease importing pork and pork products from Poland and Hungary to reduce the risk of ASFV introduction into Vietnam on 20 September 2018 [44]. However, ASFV-infected pork and pork products imported into Vietnam from Poland and Hungary before 20 September 2018 might have already been distributed within Vietnam. In the early stages of the ASF outbreak, all cases occurred on backyard pig farms, where animals were primarily fed with leftover human food containing pork products that might have been imported from Hungary or Poland without adequate treatment [45]. Because ingesting human food scraps or swill is an important pathway for ASFV transmission, it likely greatly impacted the ASFV outbreak in Vietnam. These results also support the notion that ASFV has been prevalent in Vietnam since 2019 and may have been spread through the Chinese border and Europe. Thus, our data suggest that the Vietnamese ASFV strains originated from Eastern Europe, considering the close relationship between European and Chinese strains. Moreover, Chinese ASFV strains were reported to have a close genetic similarity with European ASFV strains, suggesting that the ASFV in China and potentially Vietnam may have originated in European countries [35,36,46]. However, further genetic studies and epidemiological investigations on the origin of ASFV variants in Vietnam are required to provide concrete evidence supporting our conclusion and understanding of ASF’s transmission dynamics.

In conclusion, the whole-genome analysis of ASFVs in Vietnam during the first outbreak in 2019 provides important insights into the origin and evolution of the virus in the country. The genetic analysis showed that the ASFV strains circulating in Vietnam were closely related to strains previously detected in Estonia, China, and Poland, suggesting that the virus had likely originated from one of these countries. Our data also showed that the ASFV strains in Vietnam are undergoing rapid evolution and acquiring mutations in genomes involved in virulence or transmission, highlighting the need for continued surveillance of ASFV in Vietnam and other countries to track the virus’s evolution and prevent further outbreaks.

## Figures and Tables

**Figure 1 viruses-15-01945-f001:**
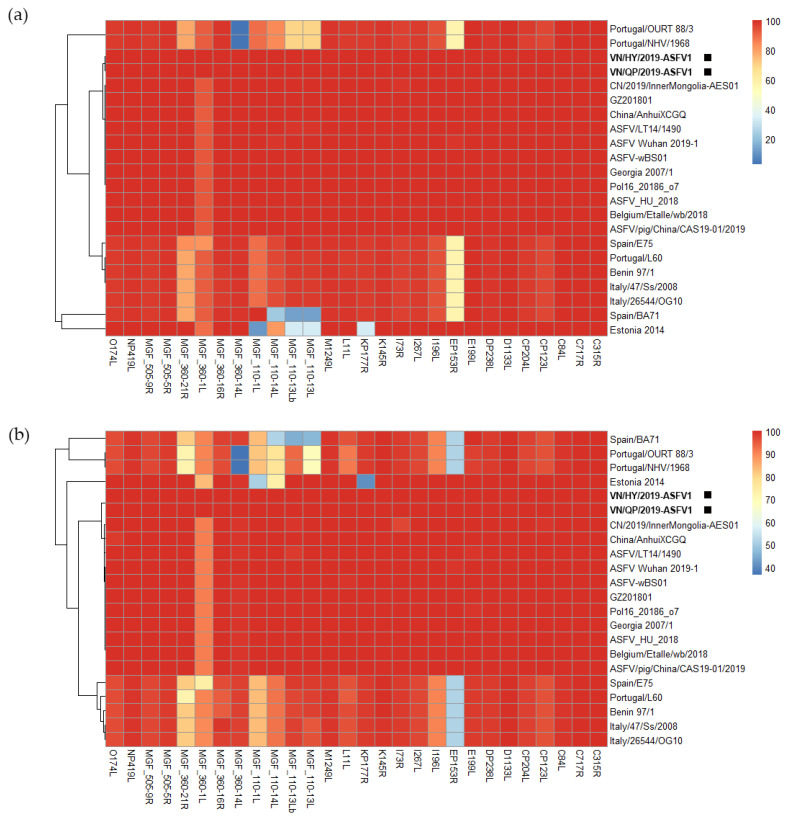
Degree of genetic variations in the conservative genes of African swine fever virus (ASFV) isolates. (**a**) Heatmap and respective dendrogram of each ASFV isolate showing the relationships between the nucleotide sequences of viral strains. (**b**) Heatmap and respective dendrogram of each ASFV isolate showing the relationships between the protein sequences of viral strains. Black squares indicate the Vietnamese strains of VN/HY/2019-ASFV1 and VN/QP/2019-ASFV1.

**Figure 2 viruses-15-01945-f002:**
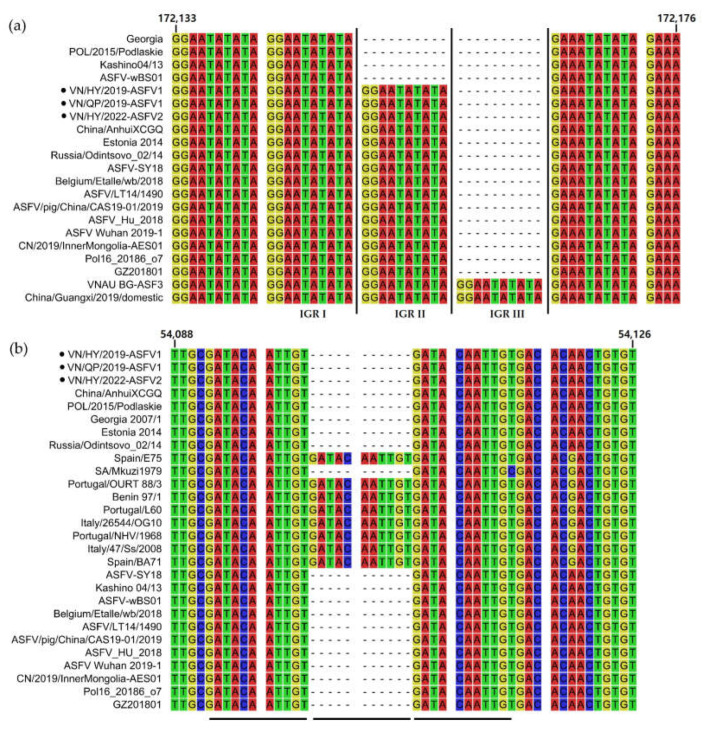
Sequence alignment of the intergenic regions (IGRs) of the African swine fever virus (ASFV) strains. (**a**) Alignment of the IGR between the I73R and I329L genes of ASFV isolates. (**b**) Alignment of the IGR between the A179L and A137R genes of ASFV isolates. Black circles indicate the VN/HY/2019-ASFV1, VN/QP/2019-ASFV1, and VN/HY/2022-ASFV2 strains analyzed in this study. Numbers above the alignment represent the positions relative to VN/HY/2019-ASFV1 (NCBI GenBank No. MT872723), and black bars indicate nucleotide tandem repeats in the IGRs. Different background colors represent adenine (A, red), thymine (T, green), cytosine (C, blue), and guanine (G, yellow), respectively.

**Figure 3 viruses-15-01945-f003:**
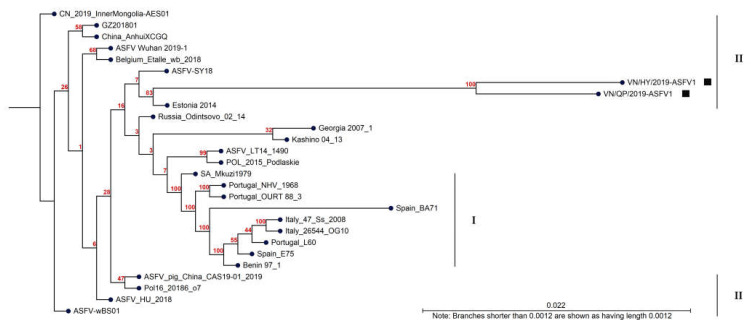
Phylogenetic relationships are based on the whole-genome sequences of the 27 African swine fever virus (ASFV) strains identified from different countries and years. The tree was constructed using the CLC Genomics Workbench and the neighbor-joining algorithm. Black squares indicate the VN/HY/2019-ASFV1 and VN/QP/2019-ASFV1 strains, and the black bars on the right indicate the p72 genotypes. The number along the branches represents 100 bootstrap values.

## Data Availability

The dataset used for this study is available within the article, and the complete genome sequences obtained were deposited in GenBank under the accession numbers MT872723, MT882025, and OR227304.

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
