# Peer review of "A Whole-Genome Analysis of the African Swine Fever Virus That Circulated during the First Outbreak in Vietnam in 2019 and Subsequently in 2022"

_viruses, 2023, doi:10.3390/v15091945_

Round 1
Reviewer 1 Report
The manuscript submitted by Jeong et al. entitled "Whole-genome analysis of the African swine fever virus that circulated during the first outbreak in Vietnam in 2019 and subsequently in 2022" analyzed the first outbreak cases of Vietnamese ASFV strains circulating in February and May 2019, and later in November 2022, and compare them with the previous circulating strains in China. The results reported provide important insights into the evolution of ASFV following its transboundary spread and genetic information for developing vaccines, and diagnostic markers to prevent the spread of ASF. The manuscript is well-written and the results are well-presented, discussed and relevant to the field.
In the opinion of this reviewer only to minor points should be improved before the publication of this work:
In line 68 a recent review about ASFV vaccines should be added (doi: 10.1080/22221751.2022.2108342) and in line 78 a more accurate epidemiologic status of ASFV in Europe should be presented (e.g. situation in Germany, Nederlands, and in Portugal no recent cases were reported!!!).
Author Response
Dear reviewers and editorial staffs in Viruses
We are sincerely grateful for your thorough consideration and scrutiny of our manuscript, “Whole-genome analysis of the African swine fever virus that circulated during the first outbreak in Vietnam in 2019 and subsequently in 2022”, control number viruses-2626587. Through the accurate comments made by the reviewers, we better understand the critical issues in this paper. We have revised the manuscript according to the Reviewer’s suggestions. We hope that our revised manuscript will be considered and accepted for publication in Viruses. We acknowledge that the scrutinizing efforts of the reviewers and editors improved the scientific quality of our manuscript.
The changes within the revised manuscript were highlighted (underlined and in blue). Point-by-point responses to the reviewers’ comments are provided below.
Reviewer #1 :
1)Reviewer’s comment: The manuscript submitted by Jeong et al. entitled "Whole-genome analysis of the African swine fever virus that circulated during the first outbreak in Vietnam in 2019 and subsequently in 2022" analyzed the first outbreak cases of Vietnamese ASFV strains circulating in February and May 2019, and later in November 2022, and compare them with the previous circulating strains in China. The results reported provide important insights into the evolution of ASFV following its transboundary spread and genetic information for developing vaccines, and diagnostic markers to prevent the spread of ASF. The manuscript is well-written and the results are well-presented, discussed and relevant to the field
In the opinion of this reviewer only to minor points should be improved before the publication of this work:
In line 68 a recent review about ASFV vaccines should be added (doi: 10.1080/22221751.2022.2108342) and in line 78 a more accurate epidemiologic status of ASFV in Europe should be presented (e.g. situation in Germany, Nederlands, and in Portugal no recent cases were reported!!!).
Author’s response: We appreciate the reviewer’s comment. As the reviewer recommended, we have replaced the reference(doi: 10.1080/22221751.2022.2108342) with [2] in the references section, and the revised introduction section is shown below:
“As only a limited number of live attenuated ASF vaccines have been commercialized and are not yet widely available, the effective prevention of ASF relies on the culling of pigs and regulation of internationally traded pork products through early detection of ASFV spread into new countries [2].”
- Urbano, A.C.; Ferreira, F. African swine fever control and prevention: An update on vaccine development. Microbes Infect.2022, 11, 2021–2033. DOI:10.1080/22221751.2022.2108342.
2) Reviewer’s comment: In line 78 a more accurate epidemiologic status of ASFV in Europe should be presented (e.g. situation in Germany, Nederlands, and in Portugal no recent cases were reported!!!).
Author’s response: We appreciate the reviewer’s comment and wholly agree with the reviewer’s opinion. We added the recent epidemiologic status of ASFV in Europe in the introduction section, and the revised introduction section is shown below:
“Recently, many ASF cases in Europe have been consistently reported in Poland, Portugal, Belgium, and Italy between 2022 and 2023. In 2023, over 100,000 cases of ASFV were reported in Poland, and over 1 million pigs were slaughtered. ASFV was reintroduced in Portugal in 2022, and over 1,000 cases have been reported since then. ASFV was first reported in Belgium in December 2022; more than 100 cases of ASFV were reported in wild boars in several country regions. No cases of ASFV were recently reported in domestic pigs in Portugal and Belgium. Moreover, ASFV was first reported in Italy in January 2023; more than 50 cases were reported, and the ASF was still spreading[39].”
Reviewer 2 Report
Since its arrival in Georgia in 2007, African swine fever has spread rapidly, through Russia and Europe, and more recently to America. Although the virus is harmless to humans, the current outbreak of African swine fever could have serious global repercussions for food security and economic stability. China – home to half the world’s pig population – has lost a third of its pigs to the disease, possibly more. The disease is spreading over more and more areas, including in asia, where it affects many countries. The disease is carried by pigs, wild boar and soft-bodied ticks, and there is currently no vaccine or cure. However, according to current knowledge , containment is possible through a combination of biosecurity, government and farmer action along with information and awareness from everyone. That's why ASFV-related studies are highly desirable and important.
The authors of this important and well-written paper have analyzed and compared the genomic sequences of ASF viruses (ASFVs) during the first outbreak in Hung Yen and Quynh Phu Province, in Vietnam in 2019, and the subsequent outbreak in Hung Yen in 2022, to those of other ASFV strains. Phylogenetic analysis based on the whole genomes of 27 ASFV strains indicated that the Vietnamese ASFV strains are genetically related to Estonia 2014, ASFV-SY18, and Russia/Odintsovo_02/14. Genetic analysis showed that ASFV strains circulating in Vietnam were closely related to strains detected in Estonia, China and Poland, suggesting that the virus probably originated from one of these countries. The results also showed that ASFV strains in Vietnam undergo rapid evolution and acquire mutations in genomes involved in virulence and transmission
Contents of all sections are appropriate and adequate. Materials and methods used in the study are adequately and fine described . Results are well presented in the manuscript as well as discussion which is comprehensive. Conclusions were justified by the obtained results and correspond to the aim of the study.
Author Response
Dear reviewers and editorial staff in Viruses
We are sincerely grateful for your thorough consideration and scrutiny of our manuscript, “Whole-genome analysis of the African swine fever virus that circulated during the first outbreak in Vietnam in 2019 and subsequently in 2022”, control number viruses-2626587. Through the accurate comments made by the reviewers, we better understand the critical issues in this paper. We hope that our revised manuscript will be considered and accepted for publication in Viruses. We acknowledge that the scrutinizing efforts of the reviewers and editors improved the scientific quality of our manuscript.
Author’s response: We really appreciate the reviewer’s comment. We need to continuously analyze virus mutations through genetic analysis of the ASF virus occurring in Vietnam, and we believe that this will be of great help in preventing further ASFV outbreaks.